# Rényi Entropy and Free Energy

**DOI:** 10.3390/e24050706

**Published:** 2022-05-16

**Authors:** John C. Baez

**Affiliations:** Department of Mathematics, University of California, Riverside, CA 92507, USA; baez@math.ucr.edu

**Keywords:** Rényi entropy, free energy, *q*-deformation *q*-derivative

## Abstract

The Rényi entropy is a generalization of the usual concept of entropy which depends on a parameter *q*. In fact, Rényi entropy is closely related to free energy. Suppose we start with a system in thermal equilibrium and then suddenly divide the temperature by *q*. Then the maximum amount of work the system can perform as it moves to equilibrium at the new temperature divided by the change in temperature equals the system’s Rényi entropy in its original state. This result applies to both classical and quantum systems. Mathematically, we can express this result as follows: the Rényi entropy of a system in thermal equilibrium is without the ‘$q^{-1}$-derivative’ of its free energy with respect to the temperature. This shows that Rényi entropy is a *q*-deformation of the usual concept of entropy.

## 1. Introduction

In 1960, Rényi [1] defined a generalization of Shannon entropy which depends on a parameter. If *p* is a probability distribution on a finite set, its Rényi entropy of order *q* is defined to be
(1)$$S_{q}=\frac{1}{1-q}\ln\sum_{i}p_{i}^{q}$$
where 0<q<∞. Of course, we need q≠1 to avoid dividing by zero, but L’Hôpital’s rule shows that the Rényi entropy approaches the Shannon entropy as *q* approaches one:$$\lim_{q\to 1}S_{q}=-\sum_{i}p_{i}\ln p_{i}.$$
Thus, it is customary to define $S_{1}$ to be the Shannon entropy.

While Shannon entropy has a deep relation to thermodynamics, Rényi entropy has not been completely integrated into this subject, or at least not in a well-recognized way. While many researchers have tried to *modify* statistical mechanics by changing the usual formula for entropy, so far, the most convincing uses of Rényi entropy in physics seem to involve the limiting cases $S_{0}=\lim_{q\to 0}S_{q}$ and $S_{\infty }=\lim_{q\to +\infty }S_{q}$. These are known as the ‘max-entropy’ and ‘min-entropy’, respectively, since $S_{q}$ is a decreasing function of *q*. They show up in studies on the work value of information [2] and the thermodynamic meaning of negative entropy [3]. For other interpretations of Rényi entropy, see the works of Harremöes [4], König et al. [5], and Uffink [6].

In fact, it is not necessary to modify the statistical mechanics to find a natural role for Rényi entropy in physics. Rényi entropy is closely related to the familiar concept of *free energy*,with the parameter *q* appearing as a ratio of the temperatures.

The trick is to think of the probability distribution as a Gibbs state, or the state of thermal equilibrium for some Hamiltonian at some chosen temperature, say $T_{0}$. Suppose that all the probabilities $p_{i}$ are nonzero. Then, when working in units where Boltzmann’s constant equals one, we can write
$$p_{i}=e^{-E_{i}/T_{0}}$$
for some nonnegative real numbers $E_{i}$. If we think of these numbers as the energies of microstates of some physical system, the Gibbs state of this system at temperature *T* is the probability distribution
$$\frac{e^{-E_{i}/T}}{Z(T)}$$
where *Z* is the partition function:$$Z\left(T\right)=\sum_{i\in X}e^{-E_{i}/T}$$
Since $Z(T_{0})=1$, the Gibbs state reduces to our original probability distribution *p* at this temperature.

Starting from these assumptions, the free energy
F(T)=−TlnZ(T)
is related to the Rényi entropy as follows:(2)$$F\left(T\right)=-\left(T-T_{0}\right)S_{\;T_{0}/T}$$
The proof is an easy calculation:$$S_{\;T_{0}/T}=\frac{1}{1-T_{0}/T}\ln\sum_{i}p_{i}^{T_{0}/T}=\frac{T}{T-T_{0}}\ln\sum_{i}e^{-E_{i}/T}=-\frac{F(T)}{T-T_{0}}.$$
This works for $T\neq T_{0}$, but we can use L’Hôpital’s rule to show that in the limit $T\to T_{0}$, both sides converge to the Shannon entropy $S_{1}$ of the original probability distribution *p*.

After the author noticed this result in the special case $T_{0}=1$ [7], Stacey commented that this case was already mentioned in Beck and Schlögl’s 1995 text on the thermodynamics of chaotic systems [8]. However, most people using Rényi entropy were unaware of its connection to free energy, perhaps because they work on statistical inference rather than physics [9]. Thus, the author put a version of this note on the arXiv in 2011 [10]. It has subsequently been cited 77 times, which suggests that it was indeed useful. We have therefore decided to publish it.

Shortly after the first draft of this note was released, Polettini gave a nice physical intepretation of Rényi entropy [11]. Downes then made a further generalization [12]. We explain those ideas here. The above argument concerns a system with Gibbs state $p_{i}=\exp\left(-E_{i}/T_{0}\right)$ at a chosen temperature $T_{0}$. Such a system automatically has zero free energy at this chosen temperature. Downes generalized the relation between Rényi entropy and free energy to systems whose free energy is not constrained this way. Polettini’s physical interpretation of Rényi entropy can be extended to these more general systems, and we describe this interpretation in what follows. We also explain how the Rényi entropy is a ‘*q*-deformation’ of the ordinary notion of entropy. This complements the work of Abe on another generalization of entropy: the Tsallis entropy [13].

In what follows, we work in a quantum rather than classical context, using a density matrix instead of a probability distribution. However, we can diagonalize any density matrix, and then its diagonal entries define a probability distribution. Thus, all our results apply to classical as well as quantum systems. The quantum generalization of Shannon entropy is, of course, well-known: it is the von Neumann entropy. The quantum generalization of Rényi entropy is also already known [14].

## 2. Rényi Entropy as a q-Derivative of Free Energy

Let *H* be a self-adjoint complex matrix. Thinking of *H* as the Hamiltonian of a quantum system, and no longer assuming that Boltzmann’s constant *k* equals 1, we may define the Gibbs state of this system at temperature *T* to be the density matrix
(3)$$\rho_{T}=\frac{1}{Z(T)}e^{-H/kT}$$
where the partition function
(4)$$Z\left(T\right)=\mathrm{tr}\left(e^{-E/kT}\right)$$
ensures that $\mathrm{tr}(\rho_{T})=1$. The Helmholtz free energy at temperature *T* is defined by
(5)F(T)=−kTlnZ(T).

On the other hand, for any density matrix ρ, the quantum generalization of Rényi entropy is defined by
(6)$$S_{q}\left(\rho \right)=k\;\frac{\ln\mathrm{tr}(\rho^{q})}{1-q}$$
since this formula reduces to the usual definition of Rényi entropy, Equation (Equation 1), when the probabilities $p_{i}$ are the eigenvalues of ρ and we set k=1. This formula makes sense when 0<q<∞ and q≠1, but we can define the quantum Rényi entropy as a limit in the special cases q=0,1,+∞. For q=1, this gives the usual von Neumann entropy:(7)$$S_{1}\left(\rho \right):=\lim_{q\to 1}S_{q}\left(\rho \right)=-k\;\mathrm{tr}\left(\rho \ln\rho \right).$$

Returning to our system with Gibbs state $\rho_{T}$ at temperature *T*, let us write $S_{q}\left(T\right)$ for $S_{q}\left(\rho_{T}\right)$. By computing this Rényi entropy at some temperature $T_{0}$, we find
$$\begin{array}{ccc} S_{q}\left(T_{0}\right) & = & k\;\frac{\ln\mathrm{tr}(\rho_{T_{0}}^{q})}{1-q}\\ & = & \frac{k}{1-q}\;\ln\mathrm{tr}\left(\frac{e^{-qH/T_{0}}}{Z{\left(T_{0}\right)}^{q}}\right)\\ & = & \frac{k}{1-q}\left(\ln Z\left(T_{0}/q\right)-q\ln Z\left(T_{0}\right)\right) \end{array}$$
If we define a new temperature *T* with
(8)$$q=T_{0}/T,$$
we obtain
$$\begin{array}{ccc} S_{q}\left(T_{0}\right) & = & k\;\frac{\ln Z\left(T\right)\;-\;q\ln Z\left(T_{0}\right)}{1-q}\\ & = & k\;\frac{T\ln Z\left(T\right)\;-\;T_{0}\ln Z\left(T_{0}\right)}{T-T_{0}} \end{array}$$
or, in short, the following:(9)$$S_{T_{0}/T}\left(T_{0}\right)=-\;\frac{F\left(T\right)-F\left(T_{0}\right)}{T-T_{0}}.$$

This equation is the clearest way to express the relation between Rényi entropy and free energy. In the special case where the free energy vanishes at temperature $T_{0}$, it reduces to Equation (Equation 2). In the limit $T\to T_{0}$, it reduces to
(10)$$S_{1}\left(T_{0}\right)=-{\left.\frac{dF(T)}{dT}\right|}_{T=T_{0}}.$$

Of course, it is already well-known that the von Neumann entropy is the derivative of −F with respect to the temperature. What we see now is that the Rényi entropy is the difference quotient approximating this derivative. Instead of the slope of the tangent line, it is the slope of the secant line.

In fact, we can say a bit more: the Rényi entropy is the the ‘$q^{-1}$-derivative’ of the negative free energy. For q≠1, the *q*-derivative of a function *f* is defined by
$${\left(\frac{df}{dx}\right)}_{q}=\frac{f(qx)-f(x)}{qx-x}.$$
This reduces to the ordinary derivative in the limit q→1. The $q^{-1}$-derivative is defined the same way but with $q^{-1}$ replacing *q*. Equation (Equation 9) can be rewritten more tersely using this concept:(11)$$S_{q}=-{\left(\frac{dF}{dT}\right)}_{q^{-1}}$$
Here we have made a change of variables, writing *T* for the variable called $T_{0}$ in Equation (Equation 9).

The concept of the *q*-derivative shows up in mathematics whenever we ‘*q*-deform’ familiar structures, obtaining new ones such as quantum groups. For an introduction, see the text by Cheung and Kac [15]. In some cases, *q*-deformation should be thought of as quantization, with *q* playing the role of exp(ℏ). That is definitely not the case here: the parameter *q* in our formulas is unrelated to Planck’s constant *ℏ*. Indeed, Equation (Equation 11) holds in classical as well as quantum mechanics.

What, then, is the thermodynamic meaning of Rényi entropy? This was pointed out by Polettini [11]. Start with a physical system in thermal equilibrium at some temperature. Then ‘quench’ it, suddenly dividing the temperature by *q*. The maximum amount of work the system can perform as it moves to thermal equilibrium at the new temperature divided by the change in temperature equals the system’s Rényi entropy of the order *q* in its original state. Note that this formulation even accounts for the minus sign in Equation (Equation 9), because it speaks of the work the system performs rather than the work performed with it.

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
