# Peer review of "Rényi Entropy and Free Energy"

_entropy, 2022, doi:10.3390/e24050706_

Round 1

Reviewer 1 Report

The author presents a simple and explicit connection between the Rényi entropy and the Helmholtz free energy, thus suggesting the same relevance that Shanon entropy has in thermodynamics. I consider that the manuscript is well written, scientifically sound and of interest. As the author recognize, there is a version, basically without change, of this manuscript in the arXiv preprint platform since 2011, and I consider that the manuscript could make surely its way to be published, however, I suggest to the author to thoroughly consider the following comments before recommendation regarding its publication.

Regarding the connection expressed by the author between the Rényi entropy and thermodynamics, I suggest to the author to dicuss the meaning of this conection in the context of the reference Phys. Rev. E 70, 017102 (2004), I understand that this might be out of the scope of the author manuscript, but a comment would be welcome.

It is a pity that some references of interest are contained in an internet blog, where the information refeered to is not easilly retrievable, I suggest to the author to make explicit, in a brief manner: the nice physical interpretation of Rényi entropy by Polettini and the generalization by Downes.

The natural logarithm symbol ln is missing in the third equality in the not numbered equation below line 53 (page 3).

In view of the connection revealed, from thermodynamics it is well known that for a mechanically isolated system kept at constant temperature the free energy nerver increases. However the explicit depedence on temperature of the Rényi entropy limits its applicability to this fundamental physical situation. Can the author comment on this aspect?

In my opinion a revised manuscript could have the potential to reignite the discussion on this interesting topic.

Author Response

The referee writes: "I suggest to the author to dicuss the meaning of this conection in the context of the reference Phys. Rev. E 70, 017102 (2004), I understand that this might be out of the scope of the author manuscript, but a comment would be welcome."

This is interesting, but it would take me a while before I could say anything about this, since I never thought before about the "observability" (in the loose sense) of Renyi entropy.  If it's unstable to small perturbations, this might somehow be related to how it can be expressed as the difference in free energies at two different temperatures.

The referee writes: " I suggest to the author to make explicit, in a brief manner: the nice physical interpretation of Rényi entropy by Polettini and the generalization by Downes."

Those are both discussed later in this paper.  I've added a bit to the introduction to clarify this: if you read the paper you're not missing anything.  My initial treatment was for T0 = 1 and Downes generalized it to arbitrary T0.  Polletini gave the "quenching" interpretation of Renyi entropy discussed at the end of the paper.

The referee writes: "The natural logarithm symbol ln is missing in the third equality in the not numbered equation below line 53 (page 3)."

Thanks!  Fixed.

The referee writes: "In view of the connection revealed, from thermodynamics it is well known that for a mechanically isolated system kept at constant temperature the free energy never increases. However the explicit dependence on temperature of the Rényi entropy limits its applicability to this fundamental physical situation. Can the author comment on this aspect?"

I'm afraid I don't have anything intelligent to say about this either. 

Reviewer 2 Report

The author tried to derive free energy from Renyi's entropy. The purpose of this study is clear, but it seems to be a very rudimentary computational result that uses the partition function format. The discussions presented here look like short reports and have not been fully theoretically considered. For example, it is not difficult to derive a formula similar to the formal free energy. Therefore, it must be said that the novelty of the methodology is poor. More importantly, it is unclear whether the author's free energy is related to the thermodynamic state orientation described by extensive macroscopic variables.
Furthermore, it should be addressed whether the author's free energy has the potential property to indicate the direction in which the physical system is reached. It is not clear how the F presented here has the meaning of thermodynamic potentials such as Gibbs and Helmholtz's free energy. Unless we can respond to the above comments, reviewers cannot recommend the publication of this paper.

Author Response

The referee writes "The author tried to derive free energy from Renyi's entropy". 

Actually I prove a relation expressing Renyi entropy in terms of free energy.   I note that an arbitrary density matrix can be interpreted as the Gibbs state at temperature T for some Hamiltonian H.   This allows us to define the Helmholtz free energy of the Gibbs state for any temperature in the usual way.  We can also define the Renyi entropy of this density matrix in the usual way.  I then prove a formula expressing the Renyi entropy in terms of the Helmholtz free energy.  While I do not state it as a formal theorem, the argument is mathematically rigorous.

The referee writes: "It is unclear whether the author's free energy is related to the thermodynamic state orientation described by extensive macroscopic variables."

Helmholtz free energy is a well-defined function of temperature for any Hamiltonian on a finite-dimensional Hilbert space.  This is the level of generality at which I am working.   As mentioned, this includes the case of any classical Hamiltonian on any finite set of classical states as a special case.  There is no concept of "macroscopic observable" at this level of generality.

The referee writes: "It is not clear how the F presented here has the meaning of thermodynamic potentials such as Gibbs and Helmholtz's free energy."

I am using the standard formula for the Helmholtz free energy of a quantum system described by any Hamiltonian on any finite-dimensional Hilbert space.  This formula is here, for example:

  • Wikipedia, Helmholtz free energy: relation to the canonical partition function.

where the energy levels Er are the eigenvalues of the Hamiltonian, counted with multiplicity.

Reviewer 3 Report

This paper studies connection between Rényi entropy and thermodynamics. Though Rényi entropy was originally created with information theoretic applications in mind it found also various application in physics (multifractals, chaotic systems, …) and estimation theory. The connection with thermodynamics was historically firstly developed in the book of Beck and Schloegel (cited by the author), and there are also independent papers of Schloegel that predate the book.

In the present paper the author elucidates the connection between Rényi entropy and Helmholz free energy. Though this connection is not new (again Beck and Schloegel were first --- by almost 15 years prior the 2011 paper), I appreciate that the author presentation is lucid, brief and insightful. Despite the fact that the manuscript is submitted to a full-fledged journal 11 years after its first appearance on arxiv, I feel that it deserves to be published in Entropy.

Author Response

Thank you for your report.

Round 2

Reviewer 2 Report

This reviewer was embarrassed to estimate this paper. The mathematical statements presented here are basic and probably not the first report because even undergraduate students can derive them. Ideas are easy to come up with and have no novelty. A thorough list of similar reports should be given in the references, and just a dozen or so volumes are not enough. In addition, there is no discussion, and therefore, this author did not present the significance according to the scientific paper format. As the author himself shows, the critical point of this paper is "What, then, is the thermodynamic meaning of Rényi entropy? This was pointed out by Polettini [12]". This author should introduce Polettini's idea and what is the novelty of this paper. The position of this study is therefore not at all clear.
If what is shown here is not clearly defined as thermodynamic entropy or potential, it is merely a report of a rudimentary student in mathematics.

Author Response

None